# Cholesterol Activates Cyclic AMP Signaling in Metaplastic Acinar Cells

**DOI:** 10.3390/metabo11030141

**Published:** 2021-02-26

**Authors:** Francesca Grisan, Martina Spacci, Carlotta Paoli, Andrea Costamagna, Marco Fantuz, Miriam Martini, Konstantinos Lefkimmiatis, Alessandro Carrer

**Affiliations:** 1Veneto Institute of Molecular Medicine (VIMM), Via Orus 2, 35129 Padova, Italy; francesca.grisan@gmail.com (F.G.); martina.spacci@studenti.unipd.it (M.S.); carlotta.paoli@studenti.unipd.it (C.P.); marco.fantuz@vimm.it (M.F.); 2Department of Biology, University of Padova, 35121 Padova, Italy; 3Molecular Biotechnology Center (MBC), Department of Molecular Biotechnology and Health Sciences, University of Torino, 10124 Turin, Italy; a.costamagna@unito.it (A.C.); miriam.martini@unito.it (M.M.); 4Department of Molecular Medicine, University of Pavia, 27100 Pavia, Italy

**Keywords:** cyclic adenosine monophosphate (cAMP), cholesterol, protein kinase A (PKA), acinar-to-ductal metaplasia (ADM)

## Abstract

Cholesterol is a non-essential metabolite that exerts both structural and signaling functions. However, cholesterol biosynthesis is elevated, and actively supports, pancreatic carcinogenesis. Our previous work showed that statins block the reprogramming of mutant KRAS-expressing acinar cells, that spontaneously undergo a metaplastic event termed acinar-to-ductal metaplasia (ADM) to initiate carcinogenesis. Here we tested the impact of cholesterol supplementation on isolated primary wild-type acinar cells and observed enhanced ductal transdifferentiation, associated with generation of the second messenger cyclic adenosine monophosphate (cAMP) and the induction of downstream protein kinase A (PKA). Inhibition of PKA suppresses cholesterol-induced ADM ex vivo. Live imaging using fluorescent biosensors dissected the temporal and spatial dynamics of PKA activation upon cholesterol addition and showed uneven activation both in the cytosol and on the outer mitochondrial membrane of primary pancreatic acinar cells. The ability of cholesterol to activate cAMP signaling is lost in tumor cells. Qualitative examination of multiple normal and transformed cell lines supports the notion that the cAMP/PKA axis plays different roles during multi-step pancreatic carcinogenesis. Collectively, our findings describe the impact of cholesterol availability on the cyclic AMP/PKA axis and plasticity of pancreatic acinar cells.

## 1. Introduction

Pancreatic adenocarcinoma (PDA) is the third-leading cause of cancer-related deaths worldwide and a major public health challenge, due to the lack of effective therapies and typical late diagnosis [1,2]. Notwithstanding the significant genetic, histological, and clinical heterogeneity of the malignancy, most cases are thought to originate from acinar cells [3,4,5]. Upon oncogenic transformation, tumors develop from normal epithelial tissue through a multi-step carcinogenic process that is determined by the accrual of genetic mutations and marked by the stepwise progression of non-invasive precursor lesions, most frequently pancreatic intraepithelial neoplasia (PanIN) but also more benign cystic neoplasms [6,7,8].

Pancreatic acinar cells are exocrine cells that aggregate in small secretory units named acini, that form about 90% of the pancreatic parenchyma [4]. Acini are accessory glands to the gastrointestinal tract, where a wide set of digestive enzymes is synthesized and stored within acidic granules (zymogens) [9]. Functional specification is supported by a unique metabolic profile, characterized by prominent amino acid catabolism [10] and sustained autophagic flux [11]. Similarly, the release of zymogens is coordinated by a dedicated Ca^2+^-dependent machinery that responds to elevation of serum levels of cholecystokinin (CCK) or direct vagal stimulation (acetylcholine) [12]. In addition, exocytosis can be stimulated by gut-derived secretin or vasoactive intestinal peptide (VIP) through signaling pathways mediated by cyclic AMP (cAMP) [13].

Pancreatic epithelium is a mixed tissue largely composed of acini, Langerhans’ islets (in turn an assortment of neuroendocrine cell types) and ducts, which collect the exocrine juice and form an anastomosis with the acini [4]. Acinar cells are intrinsically plastic and show the ability to transdifferentiate into other pancreatic epithelial lineages in adulthood [14,15]. In this respect, ductal-like cells can arise from metaplasia of the acini [16]. Indeed, acinar-to-ductal metaplasia (ADM) supports tissue homeostasis and repair after injury [17,18] and foci of ADM are normally detected in post-mortem histological examinations [19]. Oncogenic mutations of KRAS are strikingly prevalent in PDA and almost universally detected throughout the course of tumor development and already present in very early lesions [6]. Compelling evidence shows that mutant KRAS triggers ADM and makes it irreversible, an event that eventually initiates multi-step pancreatic carcinogenesis [3,4]. The way oncogenes hijack acinar cell plasticity is only partially understood; mechanisms involve extensive metabolic and epigenetic reprogramming [17,20,21] and cooption of different signaling pathways [20,21,22]. Interestingly, the three events can be intertwined [23]. While dissecting signaling/metabolic relationships in ADM, we demonstrated that metaplastic acinar cells have elevated acetyl-Coenzyme A (acetyl-CoA) levels that feed de novo sterol biosynthesis [16]. We and others also demonstrated that cholesterol-lowering drugs (statins) block in vitro ADM and suppress pancreatic tumorigenesis in preclinical models [17,24,25]. To the same point, statins usage is associated with a decreased risk for pancreatic cancer in multiple epidemiological cohorts [24,26,27]. Finally, cholesterol-rich diets that abound in western countries are linked to the onset of several types of cancer, including PDA [25]. Altogether, this evidence suggests that elevated cholesterol levels may be tumorigenic but mechanisms remain elusive. 

Administration of cholesterol to hepatocytes activates soluble adenylyl cyclase (sAC) [28] the only non-membrane-bound cyclase, which produces the second messenger cAMP [29]. In this context, internalization of circulating cholesterol triggers a cAMP-mediated cascade that leads to transcriptional rewiring and ultimately promotes fibrotic steatohepatitis [28]. While membrane-bound ACs are typically activated by ligand binding to G-protein-coupled receptors (GPCRs) [30], cholesterol acts in a receptor-independent manner and requires mobilization from the plasma membrane to intracellular compartments [28]. Whether cholesterol availability can impact GPCR-independent cAMP signaling pathways in other cell types is however not known.

The main effectors of cAMP are Protein Kinase A (PKA), guanine-nucleotide exchange factors EPAC1/2, and few cyclic nucleotide-gated ion channels [31]. The cAMP-PKA signaling axis is the most studied and best characterized [31], including for its role in acinar cell function [32], but its contribution to pancreatic carcinogenesis remains controversial [33]. Oncogenic mutations and amplifications of the GPCR-associated Gα stimulatory subunit (*GNAS*), which constitutively increase AC activity, are frequent in pancreatic cancer (2–11%) [34,35]. In mouse models of pancreatic carcinogenesis, *Gnas^R201C^* activating mutations cooperate with mutant *Kras* to drive the formation of precursor lesions [36,37]. Similarly, hereditary nonsense mutations in the PKA regulatory subunit alpha (*PRKAR1A*; mutations denote the Carney syndrome) that result in higher PKA activity, are associated with increased prevalence of pancreatic tumors [38]. On the other hand, gain-of-function mutations of GNAS are oncogenic but linked to the development of mostly benign lesions of the pancreas, which rarely evolve to carcinoma [33,36,37,39].In addition, cAMP signaling frequently promotes cell differentiation and maintenance of tissue homeostasis, while its role in cancer is more convoluted [40,41,42,43]. 

Here, we investigated the impact of exogenous administration of cholesterol on multiple signaling pathways known to be involved in acinar cell plasticity. Cholesterol supplementation to primary pancreatic acinar cells failed to potentiate signaling through epidermal growth factor receptor (EGFR) or sonic hedgehog (SHH) receptor, but rapidly augmented cAMP levels and drove PKA activation. Importantly, inhibition of cAMP-PKA signaling inhibited cholesterol-induced acinar-to-ductal metaplasia ex vivo. We observed cholesterol-triggered PKA induction in real-time using fluorescence resonance energy transfer (FRET)-based biosensors that also allow sub-cellular resolution, monitoring selected compartments. Finally, our analysis suggests that PKA regulation by cholesterol might be restricted to non-transformed pancreatic epithelial cells.

## 2. Results

We previously showed that increased cholesterol biosynthesis is critical for ductal transdifferentiation of pancreatic acinar cells [16]. To test whether exogenous administration of cholesterol can also influence cell plasticity and possibly tumor initiation, we cultured primary murine acinar explants in a synthetic matrix scaffold and ductal metaplasia was induced by the addition of recombinant EGF [4]. At the same time, cells were administered free cholesterol or vehicle, while morphology was examined after 5 days. We found that acini supplemented with cholesterol formed significantly more duct-like structures (Figure 1A).

Next, we interrogated signaling pathways known to be involved in acinar-to-ductal metaplasia (ADM) and potentially regulated by cholesterol supplementation.

Cholesterol-rich lipid rafts facilitate EGFR recycling and potentiate downstream signaling, which is critical for ADM [44]. We hypothesized that addition of exogenous cholesterol could promote the EGF-induced signaling cascade; we therefore tested the activation of critical nodes, including AKT serine/threonine kinase 1 (AKT), extracellular signal-regulated kinases-1/2 (ERK1/2) and ribosomal protein S6 quickly after stimulation with recombinant EGF. However, cholesterol did not impact phosphorylation of EGFR itself (EGFR-phospho Y1068) when provided to primary acinar cells (Figure 1B). EGF treatment could also markedly induce phosphorylation of AKT, ERK and S6, but no additive effect from cholesterol supplementation could be noted (Figure 1B).

Cholesterol covalently binds sonic hedgehog (SHH), which coordinates neighboring cell communication and embryonic development in collaboration with its receptors Patched-1/2 (PTCH1/2), Smoothened (SMO) and their second messengers GLI1/2/3 [45]. In fact, resident stromal cells support ADM through hedgehog signaling [46], but autocrine effects on acinar cells have not been described. To test the contribution of hedgehog signaling to cholesterol-induced ADM we treated primary acinar explants with the specific inhibitor Cyclopamine [47]. Morphological examination showed that Cyclopamine does not prevent duct formation upon cholesterol addition (Figure 1C, quantified by a blinded investigator in 1D). In addition, cholesterol administration does not alter the expression of *Ptch-1* (Figure 1E), which is typically negatively regulated by hedgehog effectors, while *Ptch-2* and *Gli1* are negligibly expressed in isolated acinar cells (data not shown).

Together these data demonstrate that exogenous cholesterol facilitates pancreatic acinar cell plasticity through EGFR- and hedgehog-independent mechanisms.

### 2.1. Cholesterol Activates cAMP-PKA Signaling

In vitro and in vivo models of liver steatosis showed that environmental cholesterol activates intracellular soluble Adenylyl Cyclase (sAC) in a GPCR-independent manner [28]. Acinar cells express both transmembrane (tm) and soluble (s) isoforms of AC [13,32], both playing predominant roles in zymogen secretion. AC hyperactivation also contributes to tumorigenesis in the context of oncogenic GNAS mutations [36] although the roles of cAMP-dependent signaling in pancreatic cancer are poorly understood.

We sought to demonstrate whether cholesterol administration elevates cAMP-mediated signaling in pancreatic acinar cells. To this goal, we treated primary acinar cells with cholesterol-loaded liposomes (MCD-Chol) or with Forskolin (FSK), a well-characterized activator of ACs here acting as positive control. Both treatments significantly augmented cAMP levels, although to a different extent (Figure 2A). This indicates a sub-maximal induction by cholesterol, in line with the notion that it is known to activate the relatively small sAC subset while FSK targets all but one of the transmembrane ACs [48].

Cellular levels of cAMP control the activity of Protein Kinase A (PKA), which phosphorylates numerous enzymes and eventually impacts several cellular functions [31]. Indeed, incubation with increasing doses of MCD-Chol induces phosphorylation of a broad spectrum of PKA targets as could be observed by western blotting using an antibody that recognizes substrates phosphorylated by PKA (Figure 2B). These experiments suggest that cholesterol availability elevates PKA signaling in pancreatic acinar cells. Of note, cholesterol-mediated PKA activation appears significantly lower compared to levels observed after FSK treatment, suggesting a sub-maximal stimulation by exogenous cholesterol. On the other hand, treatment with the miscellaneous PKA inhibitor N-[2-p-bromocinnamylamino-ethyl]-5-isoquinolinesulfonamide (H89) shows levels of PKA-dependent phosphorylation equivalent to cholesterol-depleted cells (Figure 2B, lanes #1 and #6); this observation indicates a poor basal PKA activity in acinar cells deprived of pre-synthesized cholesterol. Interestingly, this aligns with limited de novo synthesis of cholesterol in wild-type acinar cells as proposed in previous reports [16,49]. In line with this proposition, increasing intracellular cholesterol availability by inhibiting cholesterol esterification with an Acyl-CoA:cholesterol acyltransferase (ACAT) inhibitor does not elevate PKA activity (Appendix A).

In a similar set-up, we found that incubation with small amounts (1 µg/mL) of free cholesterol acutely activates PKA, as shown by western blotting (Figure 2C). However, longer incubation times did not show sustained PKA activity (Figure 2C), suggesting a rapid mechanism of activation that can be terminated within few hours.

Finally, to assess the role of cholesterol-mediated PKA activation in pancreatic tumorigenesis we investigated the possibility that this metabolic/signaling axis may impact acinar cell plasticity and ductal transdifferentiation. We performed morphological examination of matrix-embedded primary acinar cells. Despite the addition of recombinant EGF, wild-type acinar cells showed minimal formation of duct-like structures after 5 days (Figure 2D, quantified in E). Supplementation of free cholesterol significantly increased the number of ducts formations, while this effect was completely blunted by the addition of H89 (Figure 2D, quantified in E).

Collectively, these data show that administration of cholesterol (both free and liposome-embedded) can promote cAMP-PKA signaling axis and acinar-to-ductal metaplasia in vitro.

### 2.2. Dynamics of Cholesterol-Induced PKA Activation

To expand our findings that cholesterol provision acutely activates PKA, we deployed genetically-encoded probes based on Fluorescence/Förster Resonance Energy Transfer (FRET) that allow real-time analysis of PKA activity with superior specificity and precise spatial and temporal resolution [50]. Primary pancreatic acinar cells were isolated from 8-week-old wild-type mice and immediately infected with an adenoviral vector encoding the AKAR4 sensor [50]. Similarly, we also tested a variant of AKAR4 that specifically targets the sensor to the outer mitochondrial membrane (AKAR4-OMM [51,52]). For imaging, acinar spheroids were embedded in a synthetic matrix and fluorescence emission (which indicates PKA activity) was monitored upon addition of cholesterol in the medium. Interestingly, we registered a superior response by organelle-bound PKA (AKAR4-OMM) and a modest signal spread throughout the cell’s soluble fractions from the canonical PKA sensor (cyt AKAR4) (Figure 3A,B).

This observation may suggest that (*i*) PKA activation by cholesterol requires some sort of intracellular trafficking and that (*ii*) kinases that localize on the surface of intracellular organelles are more sensitive to cholesterol stimulation, or a combination of the two. The latter proposition is in line with our recent finding that PKA associated with intracellular membranes is more sensitive to small increases of cAMP due to a mechanism that involves phosphatase activity near intracellular membranes [51]. To begin the understanding of why cholesterol preferentially activates organelle-proximal PKA, we treated primary acinar cells with ddADO (2′,5′-dideoxyadenosine), a compound that preferentially inhibits trans-membrane AC isoforms (tmAC), or LRE1 (6-chloro-N4-cyclopropyl-N4-[(thiophen-3-yl)methyl]pyrimidine-2,4-diamine), an inhibitor selective for sAC relative to tmAC [28,53]. Interestingly, western blot analysis revealed that LRE1-treated cells did not respond to cholesterol supplementation, while treatment with ddADO had no effect on PKA activation (Figure 3C). This is consistent with the fact that cholesterol induces activation of an AC isoform not bound to the plasma membrane and possibly able to freely move within the cell. Altogether, these data reveal that extracellular cholesterol rapidly triggers PKA, preferentially in subcellular compartments that favors response to smaller fluctuations of cAMP.

### 2.3. PKA Is Not Sensitive to Cholesterol Stimulation in PDA Neoplastic Cells

Next, we tested responsiveness to cholesterol by pancreatic tumor cells. PDA is characterized by a multi-step carcinogenic process, and signaling events differ significantly across disease stage, including those related to the cAMP/PKA axis [36,54]. We therefore examined two in-house-generated cell lines derived from the clonal expansion of cells isolated from pancreatic neoplastic lesions in KC or KPC mice (*Pdx-Cre;[LSL-p53^fl/fl^];LSL-Kras^G12D^* [55,56]). The former rarely develop pancreatic carcinoma and mostly exhibit PanIN lesions; we reckoned KC cells as bona fide PanIN-derived cells. Western blotting and live imaging showed that cholesterol supplementation did not induce PKA activation in KPC or KC cells (Appendix A; data not shown). Our previous findings also described multi-faceted effects of cholesterol administration on pancreatic epithelial cells. Indeed, cholesterol promotes transformation in statin-treated acinar cells while halts proliferation of statin-treated PDA tumor cells [16]. Results here confirm that response to environmental cholesterol separates exocrine cells, early-stage neoplastic cells and tumor cells.

### 2.4. Baseline PKA Activity Evolves during Pancreatic Carcinogenesis

We reasoned that changes in downstream signaling machineries may explain the differences in cholesterol sensitivity we observed in cells from different oncogenic backgrounds. In fact, cumulative evidence points to a dichotomous role of cAMP signaling in pancreatic tumorigenesis [37,40,46,57]. To test the possibility that cAMP might be differentially active throughout disease progression, we again deployed FRET technology and the PKA sensor AKAR4 to qualitatively examine PKA activity in a panel of human and murine pancreatic epithelial cells, including non-transformed ductal and exocrine cells (HPDE and primary cells isolated from wild-type animals, respectively) and neoplastic cells modeling a wide spectrum of disease stages. We monitored FRET signals over time, and inferred PKA activity at baseline (R_b_, untreated cells), maximal (R_max_, cells treated with FSK and phosphodiesterases inhibitor IBMX [51]) and at minimal level (R_0_, after treatment with H89). Figure 4 shows time-lapsed analyses and points to markedly different patterns.

Wild-type acinar cells showed a unique pattern, defined by a rapid increase to maximal PKA activity following FSK/IBMX addition. However, treatment with H89 very slowly brought the AKAR4 fluorescence ratio at the R_0_ level, only after several minutes (Figure 4A). This pattern is unusual and could suggest weak phosphatase activity in pancreatic acinar cells and poor capacity to terminate cAMP/PKA signaling.

In striking difference, duct cells almost maximized their R_b_ (R_b_≈R_max_; Figure 4B). This showed that baseline PKA activity varies among normal pancreatic epithelial subtypes. In line with our hypothesis, PanIN-derived KC cells showed intermediate R_b_ (Figure 4C). Also interestingly, PDA tumor cells (KPC and Panc1) exhibited baseline PKA activity very close to R_0_, similar to the acinar cell phenotype and clearly diverging from their PanIN precursors (Figure 4D).

These data are consistent with the hypothesis that the cAMP/PKA signaling axis evolves during multi-step carcinogenesis [36].

Collectively, these results show a modulatory role of cholesterol on the cAMP-PKA signaling axis in pancreatic epithelial cells and its involvement in PDA initiation. Data also highlight multi-modal regulation during stepwise pancreatic carcinogenesis, which justifies the discrepancies previously reported for the roles of cholesterol and PKA signaling in pancreatic carcinogenesis.

## 3. Discussion

Sterol metabolism is crucial for the proliferation of cancer cells because its ramifications generate metabolites that are known to impact several functions, including biomass production (membrane cholesterol), energetics and redox balance (Coenzyme Q), protein compartmentalization and docking (dolichol) and signaling (several; e.g.,: protein modification, lipid rafts, hormones) [58]. The contribution of sterols to multi-step tumor development is increasingly appreciated [16,58,59,60,61].

Among sterol intermediates, cholesterol has a long-documented structural function, but other biological activities have scantly been reported, leading to the notion that it may be an inert terminal metabolite [57]. However, proteomic analysis unveiled that over 250 cellular proteins bind cholesterol [62] and its availability regulates critical transcription factors such as Sterol Regulatory Element Binding Protein (SREBP) and liver X receptor (LXR) [63]. Both SREBP and LXR activities promote tumor progression in several models [63]. In PDA, cholesterol biosynthesis and uptake are upregulated [16,49], while cholesterol deprivation impairs tumor differentiation [64] and targeting of the rate-limiting step in cholesterol biosynthesis with statins suppresses tumor formation and progression in preclinical models [17,24,25]. Pro-tumorigenic roles of cholesterol have been ascribed to LXR transcriptional regulation or to signaling events associated with cholesterol-rich domains of the plasma membrane known as lipid rafts [48,65,66], but other mechanisms likely contribute.

Here we built from the observation that cholesterol biosynthesis supports plasticity of pancreatic acinar cells [16] and interrogated mechanisms in a hypothesis-driven approach. First, we expanded our previous findings and found that exogenous administration of free cholesterol could induce acinar-to-ductal metaplasia (ADM) in a cell-autonomous manner. This implies that perturbations of cholesterol homeostasis and potentially circulating cholesterol can trigger cell-intrinsic mechanisms that initiate pancreatic carcinogenesis in mice. The ability of dietary cholesterol imbalances to promote PDA deserves further investigation.

On the other hand, we found no evidence that cholesterol supplementation could impact signaling through EGFR or SMO. This was surprising because both have a well-documented role in pancreatic carcinogenesis and both have affinity for free cholesterol, which regulates receptors’ activity [45]. Cholesterol availability also dictates lipid rafts dynamics, which influences EGFR liberty and recycle [67,68]. Yet, our data clearly point to the fact that other, possibly overlooked, signaling nodes are modulated by cholesterol during ADM. While other signaling nodes have been shown to be regulated by cholesterol trafficking (e.g., mTOR [69]), for the purpose of this study we focused only on canonical signal transduction pathways.

Cyclic AMP signaling is pivotal for cellular functioning and integral to acinar cells’ secretory activity [32], but its contribution to pancreatic carcinogenesis has been more debated [33]. Consistent with a recent report [28] and leveraging complementary approaches, we found that cholesterol administration quickly increases cAMP levels and activates PKA in primary acinar cells, putative cells of origins for PDA [5,17]. Together these experiments provide the first comprehensive description of cholesterol-mediated PKA activation. The influence of cholesterol availability on normal acinar cell function is an interesting issue. Gene-ablation analyses will more directly address the consequences of PKA loss for the pathophysiology of acinar cells.

A few points remain to be elucidated. In our experimentation we demonstrated that cAMP generation is acutely induced by extracellular cholesterol. Is that mediated by elevation of intracellular cholesterol availability? Does increased cholesterol biosynthesis activate PKA signaling, and if so, to what extent? Interestingly, our findings that cholesterol addition likely activates the soluble adenylyl cyclase and causes a more pronounced activation of PKA in proximity of mitochondria may suggest a compartment-restricted signaling role of cholesterol. Future experiments will address the dynamics of cholesterol trafficking to and from the plasma membrane and the impact of de novo biosynthesis inhibition on PKA activity, but results presented align with our previous findings that PKA signaling on the mitochondrial outer membrane is more sensitive to cAMP fluctuations [51]. Interestingly, selective induction of sAC may also explain the sub-maximal elevation of cAMP upon acute stimulation with cholesterol (Figure 2A). In the present study, we exclusively employed murine models, which does not warrant translation of our findings to human pathophysiology. It has been demonstrated that human and murine acinar cells exhibit different sensitivity to secretory stimuli (4).

Importantly, blockade of PKA signaling suppresses ADM in vitro. This analysis does not necessarily translate into a therapeutic approach mainly because cAMP signaling might not be an obligate pathway for acinar metaplasia, which is triggered by a combination of inputs/effectors [17] and mediated by extensive epigenetic remodeling [4,70]. In addition, PKA targeting (or circulating cholesterol) might endow non-acinar cells with tumor-promoting capabilities in vivo [71]. Yet, our data posit for the first time that cAMP signaling may support and contribute to pancreatic acinar cell plasticity. This was surprising as cAMP-mediated pathways have often been involved in cell differentiation and tissue functional specification [40,41,42,43]. At the same time, we linked the availability of environmental cholesterol to tumor-initiating potential; the association stands clear from an epidemiologic standpoint [65,66], but mechanisms have remained elusive.

Finally, our work indicates that baseline PKA activity, and its responsiveness to cholesterol, differ significantly across disease stages. This aligns with previous reports showing dichotomic roles of the cAMP signaling axis in PDA [33,36].

Collectively, we showed that environmental cholesterol affects pancreatic acinar cells and described cholesterol-induced PKA activation, which could represent a cell-intrinsic mechanism promoting acinar-to-ductal metaplasia. We propose that cholesterol availability might tune cAMP sensing mechanisms in acinar cells.

## 4. Materials and Methods

### 4.1. Antibodies and Reagents

Recombinant proteins, supplements and synthetic reagents used for cell treatments were: recombinant human EGF (Thermo Fisher, Waltham, MA, USA, #PHG0313; 100 ng/mL); cholesterol (Sigma-Aldrich, Saint Louis, MO, USA, #C8667). Forskolin (20 μM), IBMX (100 μM), Caliculin A (50 μM) were all purchased by Tocris Biosciences (Bristol, UK). Methyl-b-Cyclodextrin (SIGMA #128446-36); VitroGel Hydrogel (TheWell Bioscience, North Brunswick, NJ, USA, #TWG001). LRE1 and ddAdo (both 30 μM) were purchased from SIGMA (Saint Louis, MO, USA).

Inhibitors used were as follows: Cyclopamine (Cyclopamine Hydrate, SIGMA, #C4116), N-[2-p-bromocinnamylamino-ethyl]-5-isoquinolinesulfonamide (H89, 30 μM, Tocris Biosciences, Bristol, UK).

Antibodies used for Western blots include: pAKT-Ser473, AKT, pERK, ERK, pS6, S6, GAPDH, PKA phospho-substrate (RRXS*/T*) Cell Signaling Technologies, PKA catalytic subunit, vinculin, β-actin, tubulin (Sigma-Aldrich, Saint Louis, MO, USA). Secondary antibodies were from Thermo Fisher (Waltham, MA, USA). Blots were analyzed using a computerized blot scanner.

### 4.2. Acinar Cell Isolation, Culture and Treatment

Acinar cells were isolated as previously described [16], with minimal variations. Briefly, pancreata from 6–8-week-old mice were collected upon sacrifice, washed twice in cold hank’s balanced salt solution (HBSS, Biowest, Riverside, MO, USA, #L0607) and subsequently minced. Tissue was then digested with 1 mg of collagenase P (Roche, Basel, CH, #11215809103) in 5 mL HBSS at 37 °C for 30 min, occasionally inverting the tubes and interrupting the digestion every 10 min to mechanically disrupt the clogs by pipetting with progressively smaller pipette tubes. Reaction was stopped and tissue homogenate was washed twice with HBSS containing 5% calf serum (CS, Biowest, Riverside, MO, USA) #S0400) and then filtered through a 500 µm mesh and a 100 µm cell strainer. The flow through was carefully laid onto a HBSS +30% CS solution and centrifuged. The pellet was resuspended in Waymouth’s medium containing 10% CS (supplemented with soybean trypsin inhibitor Sigma-Aldrich, Saint Louis, MO, USA, #T9253 0.2 mg/mL) for suspension culture. Cells were plated in low-adhesion petri dishes, incubated at 37 °C and then harvested at selected time points as indicated in the Figure legends.

For matrix-embedded culture, cells were resuspended in a 1:3 media:VitroGel (growth-factor reduced, from TheWell Bioscience, North Brunswick, NJ, USA). Then, 200 µL of suspension were seeded onto a 48-well plate and incubated at 37 °C for solidification for at least 1 h. Upon VitroGel solidification, 500 µL of Waymouth’s medium, containing 5% CS, 0.1 mg/mL soybean trypsin inhibitor were added to each well. The following day, medium was replaced with Waymouth’s medium supplemented with 5% CS and 0.1 mg/mL soybean trypsin inhibitor and recombinant EGF (100 ng/mL, unless otherwise indicated), in addition to indicated inhibitors/supplements. Cells were monitored daily and images acquired at day 5 (unless differently reported in the Figure Legend) using a DMI6000B (LEICA, Wetzlar, Germany) inverted light microscope.

For morphological examination, a minimum of 50 pictures for condition/mouse prep was taken, randomized and evaluated by a blinded investigator (C.P.) that scored the number of acinar or ductal structures.

For FRET imaging, acinar clusters were infected with Ad-AKAR4 sensors (specifics below) immediately after isolation. After 12 h, acini were washed 4 times with PBS and again resuspended in Waymouth’s medium supplemented with 5% nCS and trypsin inhibitors. After one additional day, acini were pelleted and embedded in a synthetic matrix (VitroGel Hydrogel, TheWell Bioscience, North Brunswick, NJ, USA).

### 4.3. Quantitative PCR (qPCR)

RNA was isolated from triplicate wells under each condition using TRIzol (Invitrogen) and cDNA synthesized using high-capacity RNA-to-cDNA master mix (Applied Biosystems, Foster City, CA, USA), as per the kit instructions. cDNA was diluted 1:10 and used as template in the CAPITAL qPCR Green master mix (Biotech Rabbit, Berlin, Germany #BR0501701) and DNA was amplified using the ViiA-7 real-time PCR system. Fold change in expression was calculated using ΔΔCt, with indicated reference gene (18S, Actin or GAPDH) as an endogenous control. Primers are: *Ptch1_Fw* AGGCGCTAATGTTCTGACCA, *Ptch1_Rv* CCTCCTGCCAATGCATATAC.

### 4.4. Cell Lines and Culture

PDA-derived mouse primary cells were previously described [72]. They were cultured in DMEM supplemented with 10% Newborn Calf Serum (nCS) (Biowest, Riverside, MO, USA, S0750) supplemented with 6 mg/mL Glutamine, and penicillin/streptomycin mix.

Murine 266-6 cells were kindly donated by R. Chandwani lab (Weill Cornell, NY, USA) and cultured in DMEM medium with 10% nCS supplemented with Gln and pen/strep mix on gelatin-coated plates.

Human pancreatic duct epithelial cells (HPDE) were kindly donated by Paola Cappello and Francesco Novelli (University of Torino, Turin, Italy) and cultured in RPMI medium supplemented with 10% nCS (+pen/strep mix).

Panc1 were kindly provided by Ildiko Szabo (University of Padova, Padua, Italy) and cultured in DMEM medium with 10% nCS supplemented with Gln and pen/strep mix.

Cell lines in the lab have been authenticated by short tandem repeat (STR) profiling using the Geneprint^®^ 10 System (Promega, Madison, WI, USA). Data were matched against the ATCC reference database. All cell lines were matched to their identity with a score > 80%. ATCC names and numbers for the cell lines used in this study are as follows; Panc1 (ATCC# CRL-1469), HPDE (ATCC# CRL-4023), 266-6 (ATCC# CRL-2151). All cell lines were routinely monitored for and confirmed to be free of mycoplasma.

### 4.5. Protein Lysate Preparation for Western Blotting

Cells were washed once with 1X PBS and directly exposed to RIPA (1% NP-40, 0.5% deoxycholate, 0.1% SDS, 150 mM NaCl, 50 mM Tris-HCL pH 8.0, freshly added 1X protease inhibitors). Cells were lysated for 30 min in agitation at 4 °C. The samples were then pelleted for 5 min and 16,000 rcf and the supernatant was collected and quantified by BCA assay (EuroClone, Pero, Italy).

### 4.6. FRET Analysis of PKA Activity

Real-time tracking of PKA activity was performed as previously described. Cells were infected with adenoviral vectors encoding a compartment-specific PKA-dependent phosphorylation FRET sensor AKAR4 soluble or targeted to the outer mitochondrial membrane (cyto-AKAR4 and AKAR4-OMM respectively — ref PMID: 29941564). Viruses were prepared by Vector Biolabs (Malvern, PA, USA) and final titers were (3.7 × 10^10^) for Ad-AKAR4 and (3.7 × 10^10^) for Ad- AKAR4-OMM. After 16 h, cells were washed 4 times with PBS and embedded in a Vitrogel plug onto a glass coverslip. Coverslips were re-incubated for 2 h to allow jellification. For imaging, Vitrogel plugs were perfused using a homemade gravity-fed perfusion system with a velocity of 1 mL/min. Cells were bathed in Ringer’s modified buffer: NaCl 125 mM; KCL 5 mM; Na_3_PO_4_ 1 mM; MgSO_4_ 1 mM; Hepes 20 mM; glucose 5.5 mM; CaCl_2_ 1 mM; pH adjusted to 7.4 using 1 M NaOH. Experiments were performed on an Olympus IX81 (Olympus, Tokjo, Japan) inverted microscope equipped with a beam-splitter (OptoSplit II, Cairn Research, Faversham, UK) and a CCD camera (CoolSNAP HQ2, Teledyne Photometrics, Tucson, AZ, USA). An LED source excited the cyan fluorescent proteins at 430 nm; the emission fluorescence was collected for both donor and acceptor fluorophores at 480 nm, every 8–15 s. Automatic image collection and preliminary analysis were performed using MetaFluor software (Molecular Devices, San Jose, CA, USA). Raw data were transferred to Microsoft Excel for background subtraction and generation of the ratios; graphs were generated by Origin software (OriginLab, Northampton (MA), USA).

### 4.7. Immunoassay (ELISA) for cAMP

Equal numbers of primary acinar cells were treated with cholesterol-loaded liposomes (MCD-Chol) or Forskolin (FSK) as positive control for 40 min. Cyclic AMP was measured in duplicate for each condition using cyclic AMP Complete ELISA Kit KA0320 (Abnova, Taiwan) according to the manufacturer. Mean results from independent acinar cultures derived from 3 wild-type animals are presented.

### 4.8. Statistical Analysis

Data are presented as the means of experimental replicates with their respective standard deviations (SD), unless otherwise indicated. Student’s two-tailed *t*-test (two-sample equal variance, two-tailed distribution) was used for analyses, unless otherwise indicated. Significance was defined as follows: *, *p* < 0.05; **, *p* < 0.01; ***, *p* < 0.001.

## Figures and Tables

**Figure 1 metabolites-11-00141-f001:**
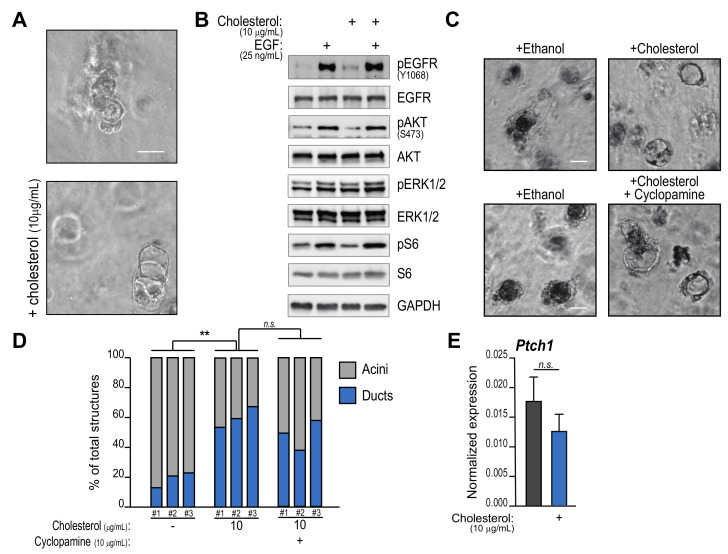
Cholesterol supplementation induces acinar-to-ductal metaplasia (ADM) but does not significantly impact EGFR or sonic hedgehog (Shh) signaling. Primary wild-type pancreatic acinar cells were either embedded in a synthetic hydrogel cast (**A**,**C**,**D**,**E**) or cultured suspended in RPMI medium (**B**) and stimulated with recombinant EGF (100 ng/mL, unless otherwise indicated). A, Cells were treated with free cholesterol (10 μg/mL) or vehicle (ethanol). After 5 days, morphology of cellular clusters was examined. Figure shows representative images of triplicate experiments. Scale bar, 50 μm. B, Cells were lysated 20 min after EGF addition. Western blotting shows phosphorylation of EGFR and downstream effectors. Representative image of duplicate experiment. C–D, Acinar cells were independently isolated from 3 mice (#1, #2, #3) and treated with cholesterol and SHH inhibitor Cyclopamine or vehicle. Morphology was examined after 5 days (C) and number of acini/duct structures was quantified by a blinded investigator (in D). **, *p* < 0.01. Scale bar, 50 μm. E, qPCR analysis of the Ptch1 gene in cells treated like in A.

**Figure 2 metabolites-11-00141-f002:**
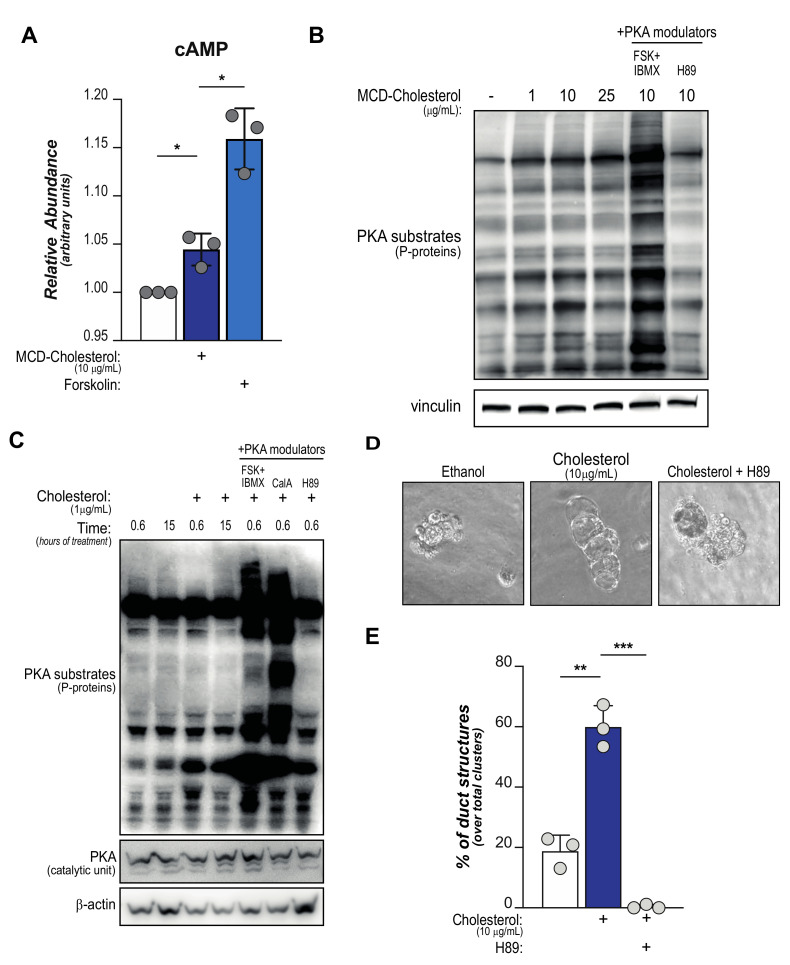
Cholesterol supplementation promotes PKA activation in pancreatic acinar cells. Pancreatic acinar cells were isolated from wild-type animals and supplemented with free- or MCD-bound-cholesterol at indicated concentrations. (**A**), cAMP levels were measured in independent acinar explants (*n* = 3). Forskolin was used as positive control. *, *p* < 0.05. (**B**,**C**), Western blotting shows PKA-specific phosphorylation sites in acinar cells treated (**B**) with increasing concentrations of cholesterol or indicated compounds for 40 min or (**C**) with minimal cholesterol for extended times (indicated). In (**C**), total PKA abundance is also shown. (**D**,**E**), Pancreatic acinar cells were isolated from wild-type mice (*n* = 3), embedded in a synthetic hydrogel cast and stimulated with recombinant EGF (100 ng/mL). Cells were treated with free cholesterol (10 mg/mL) ± H89 or vehicle. After 5 days, morphology of cellular clusters was examined. Representative images are shown in D, blinded quantification of duct-like structures in E. *, *p* < 0.05; **, *p* < 0.01; ***, *p* < 0.001. For all panels, FSK: forskolin (20 µM); CalA: caliculin-A (5 µM); H89: N-[2-p-bromocinnamylamino-ethyl]-5-isoquinolinesulfonamide (30 µM).

**Figure 3 metabolites-11-00141-f003:**
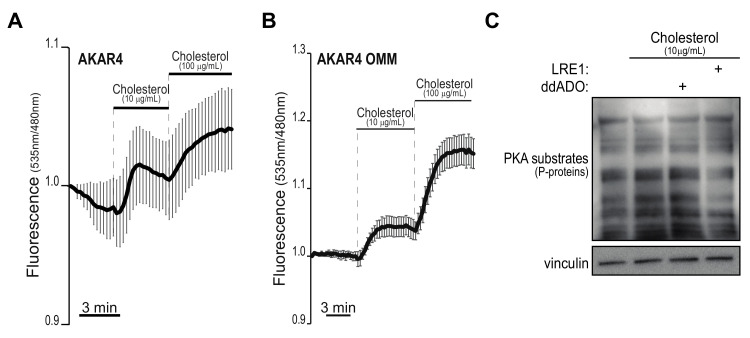
Cholesterol supplementation induces rapid PKA activity predominantly in proximity of sub-cellular organelles. (**A**,**B**), Pancreatic acinar cells were isolated from wild-type animals, infected with adenoviral vectors encoding for the PKA sensors AKAR4 (that senses cytoplasmic PKA activity, panel (**A**) and AKAR4-OMM (that targets PKA activity localized on the surface of intracellular organelles, panel (**B**). After 36 h, cells were embedded in a synthetic matrix and imaged. Fluorescence intensities were recorded for several minutes, before and after the addition of indicated amounts of cholesterol. The graphs show FRET ratios over time. Scale bars, 3 min. In (**C**), primary acinar cells were added cholesterol or vehicle and immediately treated with indicated inhibitors (30 µM both) for 15 h. Whole-cell lysates were blotted to detect phosphorylated PKA substrates.

**Figure 4 metabolites-11-00141-f004:**
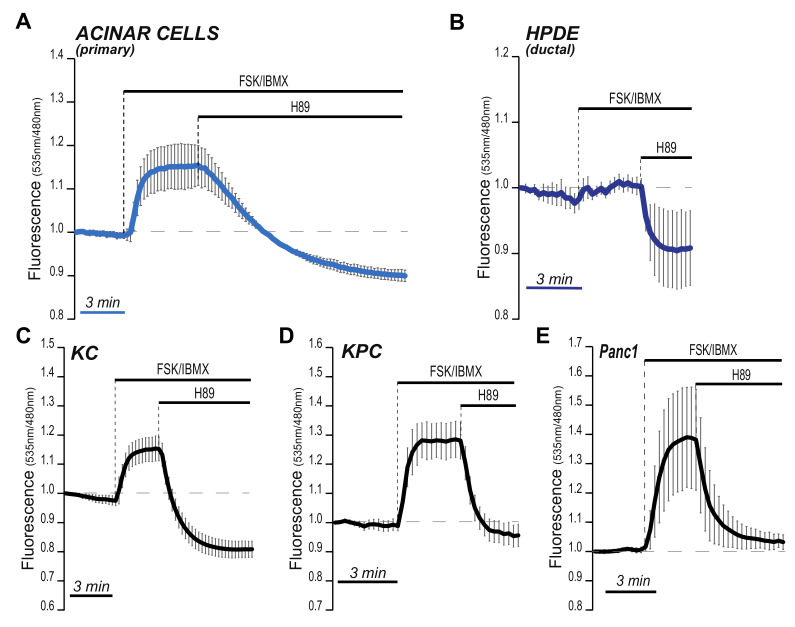
Stage-specific PKA activity in pancreatic cancer cells. Pancreatic epithelial cells (non transformed, ductal and acinar, and tumoral, indicated at the top of each panel) were infected with adenoviral vectors encoding for the PKA sensor AKAR4 and imaged after 36 h. Fluorescence intesities were recorded for several minutes, before and after the addition of indicated compounds. The graphs show FRET ratios over time. Scale bars, 3 min. In (**A**), mouse-derived primary acinar cells; in (**B**), normal duct cells (commercially-available; HPDE); in (**C**), primary mouse cells derived from bona fide pre-malignant lesions (PanIN); in (**D**,**E**), PDA tumor cells, both primary (mouse-derived) and from an established human cell line (Panc1). Graphs are representative of three independent experiments.

## Data Availability

The data presented in this study are available in this article and Appendix A.

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
