# Peer review of "Cholesterol Activates Cyclic AMP Signaling in Metaplastic Acinar Cells"

_metabolites, 2021, doi:10.3390/metabo11030141_

Round 1

Reviewer 1 Report

This is an excellent, well organized, and easy to read manuscript. It has been written with diligence and seems to be comprehensive. No criticisms from my side. I recommend publication as it is.

Author Response

We thank the Reviewer and appreciate her/his positive feedback.

Reviewer 2 Report

Grisan et al in their manuscript titled “Cholesterol activates cyclic AMP signaling in metaplastic acinar cells“ investigted the impact of exogenous administration of cholesterol on multiple signaling pathways known to be involved in acinar cell plasticity. The authors showed that cholesterol supplementation to primary pancreatic acinar cells failed to potentiate signaling through EGF receptor (EGFR) or Sonic Hedgehog (SHH) receptor, but rapidly augmented cAMP levels and drove PKA activation. Inhibition of cAMP-PKA signaling inhibited cholesterol-induced acinar-to-ductal metaplasia ex vivo. They observed cholesterol-triggered PKA induction in real-time using Fluorescence Resonance Energy Transfer (FRET)-based biosensors that also allowed them to sub-cellular resolution, monitoring selected compartments. The authors conclude that the ability of cholesterol to activate cAMP signaling is lost in tumor cells. Qualitative examination of multiple normal and transformed cell lines supports the notion that the cAMP/PKA axis plays different roles during multi-step pancreatic carcinogenesis. Collectively, our findings describe the impact of cholesterol availability on the cyclic AMP/PKA axis and plasticity of pancreatic acinar cells.

The authors have done good work, which is an extension, their previous work.

I have tow minor points.

  1. Please write gene names, an in vitro, in vivo in italics.
  2. Please tone down your conclusions with warrant to do more experiments and repetition in human models or explicitly state that these findings are predominantly murine specific.
  3. Line 419 write 40C

Reviewer 3 Report

This paper presents further data on the metabolism of cholesterol by cancer cells of pancreatic origin. It will appeal to researchers in the niche field of exploring therapeutic strategies aimed at interfering with cholesterol metabolism in cancer. It is well presented and discussion is to the point.

Only minor typos such as in Fig1 "W esten blotting"

Author Response

We thank the Reviewer and apologize for the typos, which have now been fixed. To that point, the manuscript has been proof-read by a professional editor.